# Particle-sounding of the spatial structure of kinetic Alfvén waves

Z.-Y. Liu [1], Q.-G. Zong [1,2,3] ✉, R. Rankin [4], H. Zhang [5], Y.-X. Hao [6], J.-S. He[1], S.-Y. Fu[1], H.-H. Wu[7], C. Yue [1], C. J. Pollock[8] & G. Le[9]

Kinetic Alfvén waves (KAWs) are ubiquitous throughout the plasma universe. Although they are broadly believed to provide a potential approach for energy exchange between electromagnetic fields and plasma particles, neither the detail nor the efficiency of the interactions has been well-determined yet. The primary difficulty has been the paucity of knowledge of KAWs' spatial structure in observation. Here, we apply a particle-sounding technique to Magnetospheric Multiscale mission data to quantitatively determine the perpendicular wavelength of KAWs from ion gyrophase-distribution observations. Our results show that KAWs' perpendicular wavelength is statistically $2.4 \pm 0.7$ times proton thermal gyro-radius. This observation yields an upper bound of the energy the majority proton population can reach in coherent interactions with KAWs, that is, roughly 5.76 times proton perpendicular thermal energy. Therefore, the method and results shown here provide a basis for unraveling the effects of KAWs in dissipating energy and accelerating particles in a number of astrophysical systems, e.g., planetary magnetosphere, astrophysical shocks, stellar corona and wind, and the interstellar medium.

Waves pervade the plasmas in the universe[1], mediating the transport, cascade, conversion and dissipation of the energy of various astrophysical and space objects. They are also believed to be partially, if not dominantly, responsible for the high-energy charged particles observed in space and on the ground[2]. Among various parameters, waves' spatial scale, i.e., wavelength, serves as one of the most critical quantities controlling wave dynamics. For example, it characterizes different wave modes[1,3], determines turbulent energy cascading and dissipation[4,5], and controls the efficiency of wave-particle interactions[6–8].

Determining wavelength of plasma waves directly from spacecraft in-situ observations is an indispensable step towards a general account of plasma-wave physics and their effects in dissipating energy, heating matter and accelerating particles. To this end, many methods have been proposed, which can be divided into four classes in general. The first class of methods is based on phase differences measured among spacecraft at different points in space, like the timing[9,10] and k-filtering analysis[11]. The second class of methods is also based on multi-point observations, but now the key information is spatial gradient, whose reciprocal gives wavelength provided that quasi-monochromatic waves are the only source of spatial variations[12]. The third class of methods derives wavelength indirectly from a set of variables linked by it. For example, wave number, electric current and magnetic field form a closed equation via Ampére's law; the knowledge of the latter two gives the former[12,13]. The last class of methods utilizes wave dispersion relation presenting wave frequency as a function of wave number. One may first construct or verify the dispersion relation from the observed ratio of electric fields to magnetic fields, from which wavelength can be extracted[14,15]. For the sake of completeness, we also note the wavelength can sometimes be derived directly from the spatial distributions

[1]Institute of Space Physics and Applied Technology, Peking University, Beijing, China. [2]Key laboratory of solar activity and space weather, National Space Science Center, Chinese Academy of Sciences, Beijing, China. [3]Polar Research Institute of China, Shanghai, China. [4]Department of Physics, University of Alberta, Edmonton, AB, Canada. [5]Geophysical Institute, University of Alaska Fairbanks, Fairbanks, AK, USA. [6]Helmholtz Centre Potsdam, GFZ German Research Centre for Geosciences, Potsdam, Germany. [7]School of Electronic Information, Wuhan University, Wuhan, China. [8]Denali Scientific, Fairbanks, AK, USA. [9]NASA Goddard Space Flight Center, Greenbelt, MD, USA. ✉e-mail: qgzong@pku.edu.cn

of relevant variables, if available. This method is usually applied in, for example, the study of aurora imaging[16] and laboratory experiments[17,18].

Many interesting and important results have been obtained via these methods (e.g., ref. [12]). However, the practice also shows that these methods could suffer from large errors or even cannot be applied due to their inherent limitations. First of all, in many situations multi-point observations are not available, especially in the exploration of space beyond the Earth. Second, even when multi-point observations are available, only wavelength close to spacecraft separation can be accurately determined; otherwise, significant errors arise[19]. Finally, some of the proposed methods, especially those based on single-point measurements, rely on variables whose accurate measurements are not easy to obtain. For example, spacecraft charging[20], which usually occurs due to dense plasma nearby or sunlight, would hinder the detection of thermal (a few tens of eV) electrons and ions and thus space electric current, which is required in deriving wavelength from Ampére's law.

One particular plasma wave mode for which the accurate determination of its spatial scale is important is kinetic Alfvén waves[3] (KAWs). This wave mode can be considered as an extension of magnetohydrodynamic (MHD) Alfvén-branch waves in the kinetic regime. It arises from the non-ideal effects in the generalized Ohm's law, which become significant when ion motion decouples from electrons' at $k_\perp \rho_{i,th} \sim 1$, where $k_\perp$ denotes the wavenumber perpendicular to the background magnetic fields and $\rho_{i,th}$ represents thermal proton gyroradius evaluated at their local perpendicular thermal energy. Since the details of the decoupling process are sensitive to spatial scale, determining KAWs' wavelength is a starting point for studying their properties and how they interact with particles and other plasma waves.

The scientific interest in KAWs is determined by their essential role in various plasma phenomena. They have long been proposed as an energy source of planetary aurora[21,22] and space weather-induced atmospheric escape[23]. They are also partially responsible for magnetospheric activities that make the near-Earth environment harmful to satellites and astronauts[24,25]. Additionally, they are one of the crucial ingredients composing the space and astrophysical turbulence[26,27] and thus offer a potential answer to the plasma heating puzzle of the solar corona, the solar wind, magnetospheres, and the interstellar medium. Besides natural settings, they are important for laboratory plasmas as well, especially in transporting energy in magnetic fusion devices[28]. Accurate knowledge of KAWs' spatial scale in the full parameter space could provide significant new insight for unraveling the physics of these processes.

In this paper, we propose application of an innovative particle-sounding technique for determining the wavelength of KAWs and other small scale waves from in-situ spacecraft observations of particle gyrophase distributions. The idea behind this technique is, as illustrated in Fig. 1, the gyration of charged particles of different energy in magnetic fields forms a "ruler" to measure the wavelength of nearby plasma waves. The only required inputs of this technique are single-spacecraft measurements of magnetic field and the velocity distributions of particles used to disclose the "ruler" (these particles usually have energy well above spacecraft potential, and thus can be reliably detected). With this technique, we can now measure wavelength and reveal associated physics of KAWs in situations where previous methods cannot be applied.

## Results
### Overview of the 30 December 2015 event

The KAWs studied here were observed by NASA's Magnetospheric Multiscale (MMS) spacecraft in Earth's magnetosphere on 30 December, 2015. They were first reported by Gershman et al.[12], who used four independent methods (described in the Introduction) to estimate their wavevector. The results that **k**=[0.7, −2.0, −2.2]×10$^{-2}$ rad/km in the Geocentric Solar Ecliptic or GSE coordinates indicate that the wavelength perpendicular to the background magnetic field is comparable to the gyro-radius of local thermal protons. From this, together with an analysis of the dispersion relation (their Fig. 5), Gershman et al. concluded the KAW-mode nature of the observed waves. After identifying the wave mode, Gershman et al. analyzed how electrons respond to KAWs' fields. However, they did not elucidate the ion dynamics induced by the KAWs, which was the main focus of our investigation here.

Figure 2 shows an overview of the KAW-ion interactions. During the magnetopause crossing, MMS detected a plasma jet flowing approximately anti-parallel the background magnetic field (**B$_0$** = 55[0.10, −0.52, 0.85] nT in GSE) at about 22.26 UT. The speed of the jet was about 250 km/s, corresponding to a proton kinetic energy of about 326 eV. The KAWs appear as wave packets evident in Figs. 2b–d, which is a zoom into a particular wave packet when the instruments onboard MMS were in burst mode. The frequency of the observed waves is about 0.88 Hz (Supplementary Fig. 1c) in the rest frame of the spacecraft, which is close to the proton gyrofrequency (about 0.84 Hz) derived from **B$_0$** and is at the upper end of the frequency range of KAWs. The wave fields are dominated by their perpendicular components ($B_\perp \approx 3$ nT and $E_\perp \approx 3$ mV/m) that are approximately left-hand circularly polarized (Fig. 2d where $B_{\perp 1}$ lags $B_{\perp 2}$ by about 90° in phase and Supplementary Fig. 1d). However, significant oscillations can also be observed in the parallel component of the wave electric fields ($E_\parallel$); the amplitude of $E_\parallel$ is about 0.7 mV/m, as shown in Fig. 2d. To determine wave propagation, a minimum variance analysis[29] is applied to the wave magnetic fields. The results show the waves propagate quasi-perpendicularly to **B$_0$**, with a wave normal

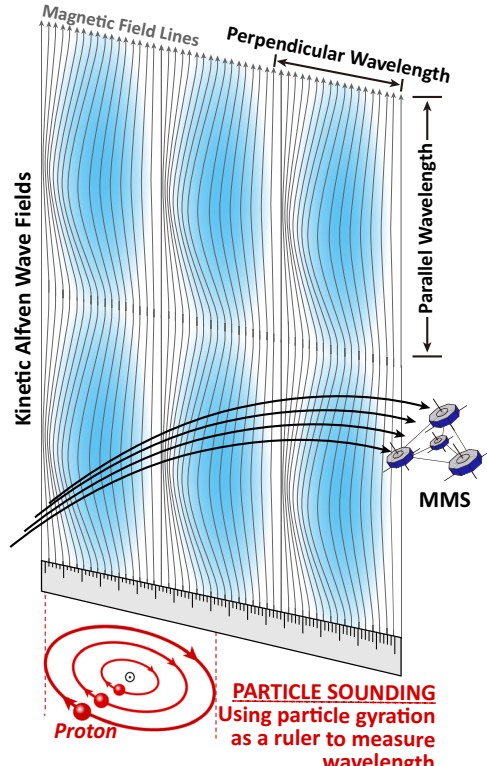

**Fig. 1 | Schematics of particle-sounding technique (not to scale).** The thin grey curves represent the magnetic field lines. The blue background shows the phase of KAWs. The red curves at the bottom illustrate the gyration of protons of different energy in the magnetic field. The interactions between KAWs and protons would significantly modulate the distribution function of protons, which can then be observed by the MMS spacecraft. This observation, in turn, provides a "ruler" to derive the perpendicular wavelength of the KAWs.

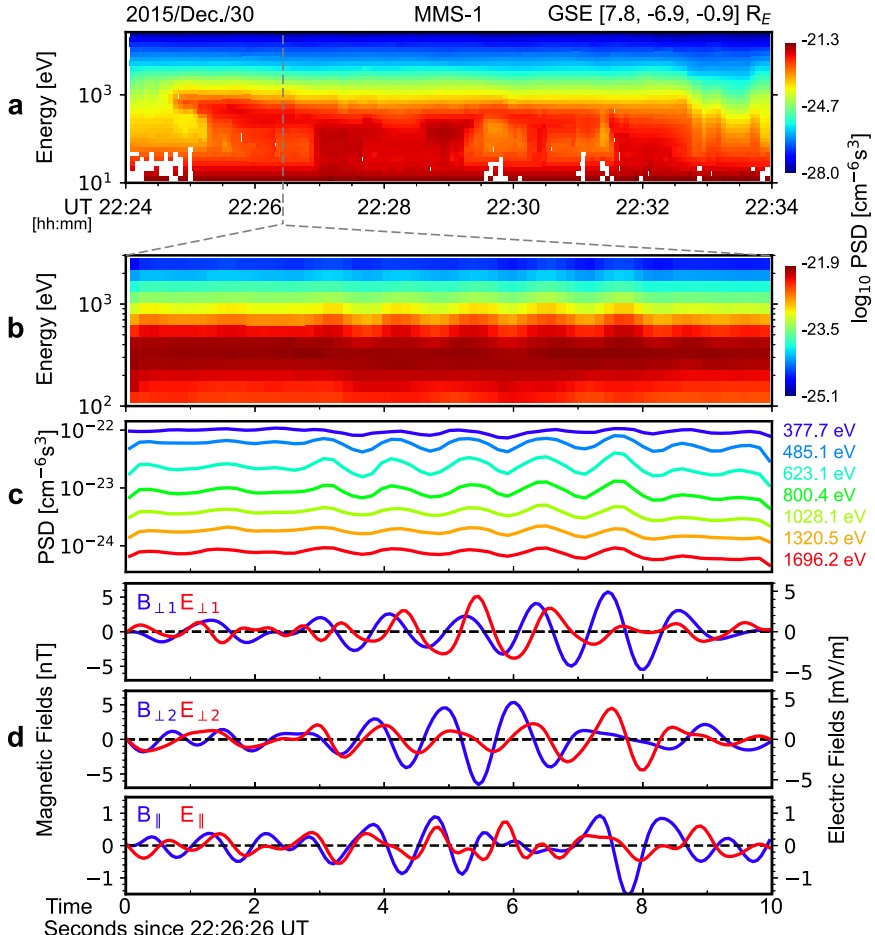

**Fig. 2 | Overview of the KAW-proton interaction event on December 30, 2015.** **a** The looking direction-averaged phase space densities of protons as a function of time and energy. **b** An expanded view of panel a, in which KAW-associated oscillations can be noted. **c** The line-version of panel b. **d** The magnetic and electric fields of the KAWs in the field-aligned coordinates.

angle of about 88.5°. The properties listed above support the identification of the waves as KAWs[30,31].

Figures 2b, c shows ion phase space densities (PSDs) as a function of time and energy. For this and the following figures, ion data are taken from the FPI instruments[32] onboard MMS (see detailed description in Methods, subsection Data Processing). Although the FPI instruments do not distinguish between different ion species, the HPCA instruments[33] on MMS, which cover a similar energy range as the FPI instruments, suggest the bulk of the ions detected by the FPI instruments are protons. Therefore, the gyro-radii of the ions detected by the FPI instruments can be calculated based on the proton mass and charge, although the multi-ion effects[34–36] on KAW physics cannot be fully ruled out. The latter effects, however, are beyond the scope of this paper, and therefore, we will hereafter refer to the FPI ion measurements as proton measurements for convenience.

As shown in Figs. 2b, c, quasi-periodic oscillations are present in proton PSDs from about 200 eV to 2000 eV. A comparison between Figs. 2c, d suggests the oscillations in proton PSDs are well correlated with the oscillations in wave fields, indicating a causal relationship between them. In what follows, we examine this relationship in more detail.

**Proton non-gyrotropic distributions induced by KAWs**

Figure 3 shows the detailed proton distribution functions observed by MMS. For reference, Fig. 3a shows a perpendicular component of the wave electric field ($E_{\perp 1}$). Figures 3b-d shows the pitch-angle time spectrograms of proton gyrophase-averaged PSDs. Superimposed on

the anti-parallel jet manifesting as PSD peaks at pitch-angle>90°, the PSDs oscillate in phase with $E_{\perp 1}$. Within 200–2000 eV where the oscillations are significant, the oscillations are much clearer at higher energies. It is important to note that this observation is not consistent with the expected behavior of pure cyclotron wave-particle interactions. As shown in ref. 37, this interaction would not cause any significant oscillations in proton gyrophase-averaged PSDs. Indeed, $E_{\parallel}$ can result in some gyrophase-averaged PSD oscillations, since $E_{\parallel}$ is capable of trapping particles at parallel potential valleys. However, this effect should manifest at lower energy, in contrast to the observations presented here. Also, the depth of the parallel potential valleys of the KAWs studied here is only -10 V[12], indicating that $E_{\parallel}$ cannot easily modulate the motion of >200 eV protons significantly and thus is not the cause of the oscillations shown in Figs. 3b-d. To explain the physical properties of the observed PSD oscillations, one should turn to other properties of KAWs.

Figures 3e–g and Supplementary Fig. 2 show proton gyrophase distributions, namely, the distribution functions in the plane perpendicular to $\mathbf{B}_0$. Here, only protons near the anti-parallel jet velocity are included in the spectrogram. Thus, protons of different energy actually imply different perpendicular velocities. In the 377.7-eV channels, more than five continuous inclined stripes are obvious. These stripes have a negative slope on the spectrogram and repeat at about 1-second intervals, following the frequency of the KAWs. Similar distributions have been reported in recent observations of cyclotron wave-ion interactions and explained as nonlinear gyrophase bunching of ions by wave perpendicular electric fields[37–39].

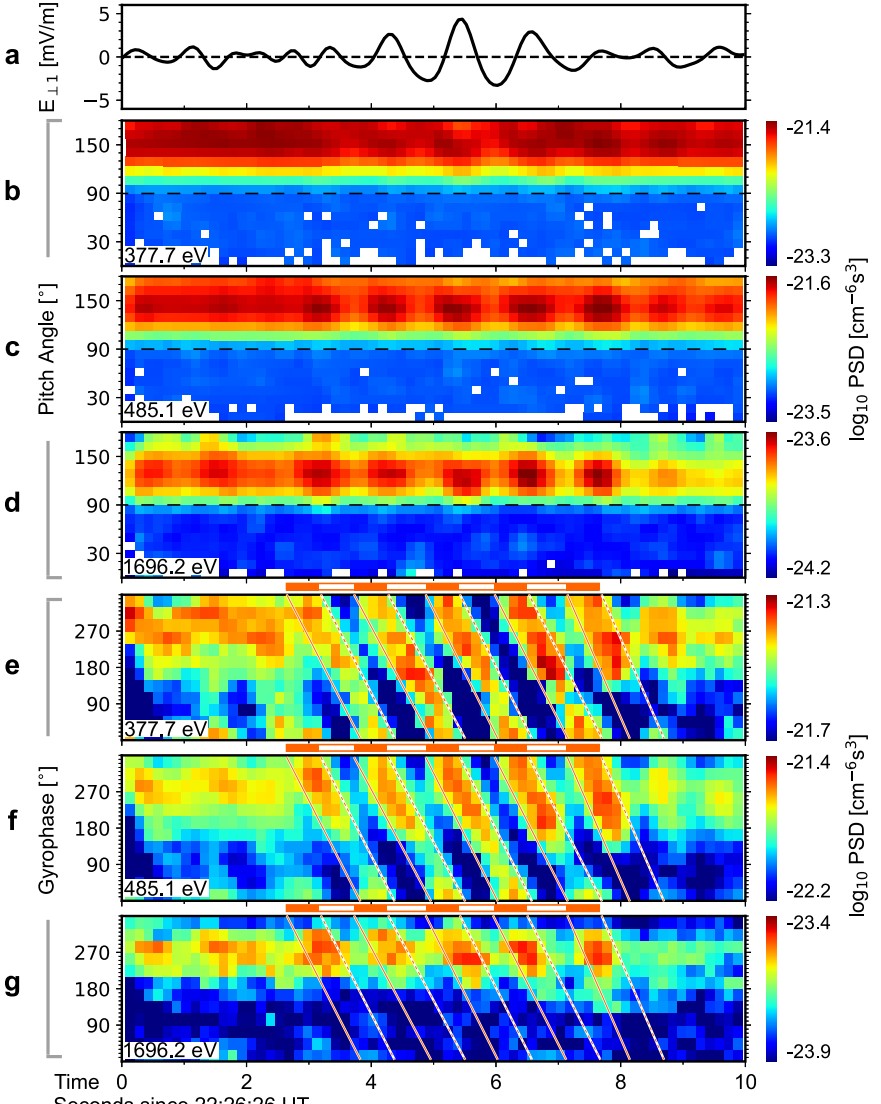

**Fig. 3 | Detailed observations of proton dynamics induced by KAWs. a** A perpendicular component of the KAW electric fields. **b–d** The pitch angle distributions of protons. Gyrophase-averaged phase space densities are shown. **e–g** The gyrophase distributions of protons with the same parallel velocity, 250 km/s. Phase space densities are shown. The orange solid and dashed lines give the expected positions of phase-bunching stripes.

Phase-bunching stripes (PBSs) are also observed in higher energy channels (Figs. 3f, g). However, while the low-energy PBSs change continuously from 0° to 360° gyrophase, the high-energy PBSs only cover a narrower gyrophase range, leading to a gap centered at about 90° gyrophase in each stripe. As shown in Fig. 3 and Supplementary Fig. 2, the gyrophase range covered by PBSs systematically decreases as proton energy increases. This decreasing tendency of PBSs with energy finally leads to the disappearance of PBSs at >2000 eV. No clear wave-associated signatures can be observed at these energies in the proton energy-, pitch angle- and gyrophase-time spectrograms, indicating full decoupling of more energetic protons from the observed KAWs. To the best of our knowledge, the energy-dependent gyrophase-coverage of PBSs is a unique feature of the KAW-ion interaction that has not previously been reported.

To quantitatively analyze the PBSs and their gaps, we first employ a superposed epoch analysis to generate superposed gyrophase-wave phase spectrograms from the gyrophase-time spectrograms (details are provided in the Methods section). The right column of Supplementary Fig. 2 gives the results. One can observe in these spectrograms again the PBS gaps discussed earlier. Then, we select PSDs along PBSs (the white lines in Supplementary

Fig. 2). Finally, we normalize the resulting PSDs by dividing them by their maximum value. Figure 4a shows the normalized PSDs as a function of energy and gyrophase. At lower energy, all gyrophases are marked by red colors, suggesting the PSDs here are approximately the same along PBSs. In contrast, at higher energy, a large extent of gyrophases are marked by blue colors representing relatively smaller PSDs, suggesting the PSDs vary significantly along the corresponding PBSs. The ratio between the red color- and blue color-marked gyrophases systematically decreases with increasing energy, as illustrated by the black curves marking the 0.8 level in Fig. 4a.

There are three properties of PBS gaps that shed light on their formation. First, the PBS gaps are centered around 90° gyrophase, which is approximately opposite the propagation direction of the KAWs in the plane perpendicular to $\mathbf{B_0}$ (about 270° gyrophase; see the wavevector $\mathbf{k}$ given above). Second, as shown on the right vertical axis of Fig. 4a, the PBS gaps occur when $k_\perp \rho_i \sim 1$ with $\rho_i$ representing proton gyro-radius at any given perpendicular energy and $k_\perp$ denoting the perpendicular component of $\mathbf{k}$. For $k_\perp \rho_i \ll 1$, continuous PBSs are observed instead. Third, besides the jet velocity, similar PBS gaps can be observed at other parallel velocities.

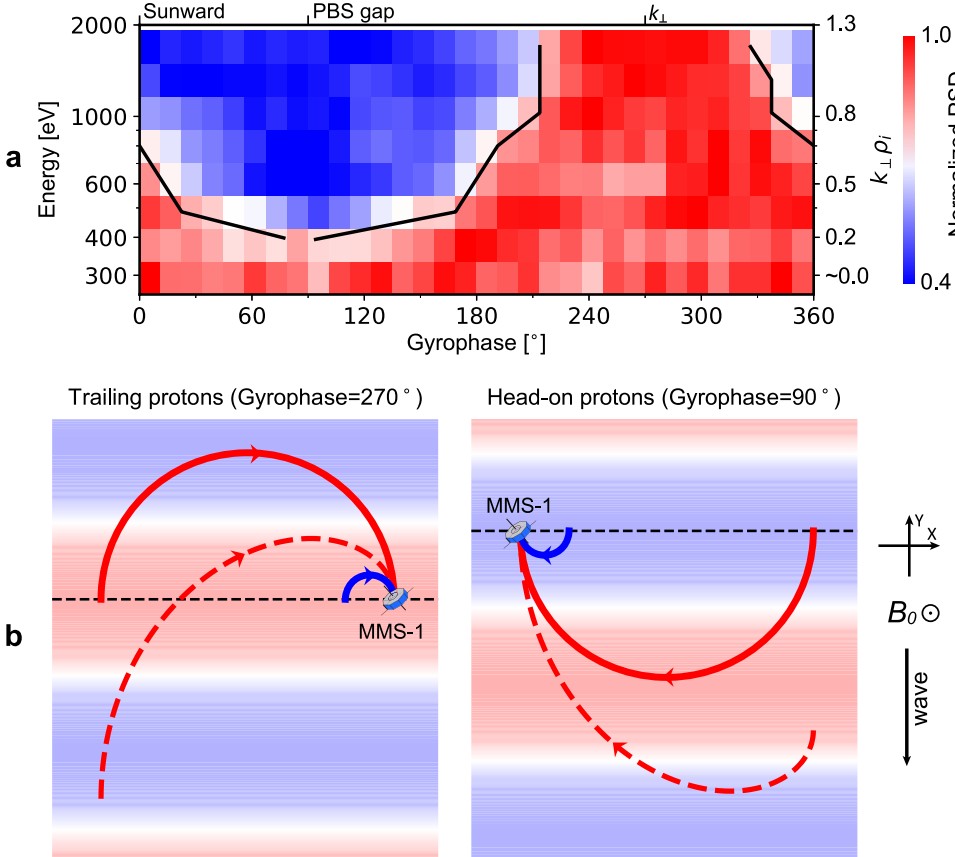

**Fig. 4 | KAW-proton interactions. a** The extent of phase-bunching stripe gaps (marked by the blue colors), which systematically increases as proton perpendicular energy and thus $k_\perp \rho_i$ increase. The black curve marks the 0.8 level. From left to right, The three labels on the top give the gyrophases of the sunward direction, PBS gaps and the propagation direction of the KAWs in the perpendicular plane. **b** Diagrams illustrating the geometry of KAW propagation and proton trajectory in the plane perpendicular to $\boldsymbol{B_0}$. The red-blue background illustrates wave phases.

The solid blue and red curves represent the trajectories of protons of small gyro-radius ($k_\perp \rho_i \ll 1$) and large gyro-radius ($k_\perp \rho_i \sim 1$), respectively, in the absence of wave perpendicular propagation. The red dashed curve represents the trajectories of protons of large gyro-radius in the presence of wave perpendicular propagation, looking in a frame of reference moving at the perpendicular phase velocity of the waves. The left and right columns show trailing protons forming 270°-gyrophase PBSs and head-on protons forming 90°-gyrophase PBSs, respectively.

Moreover, at any given value of $k_\perp \rho_i$, the width of the PBS gaps varies little with proton parallel velocity, as shown in Supplementary Fig. 3.

## KAW-proton interaction at ion scales

The occurrence of PBS gaps at $k_\perp \rho_i \sim 1$ suggests that finite Larmor radius effects are essential for the KAW-proton interactions and lead us to the following interpretation. We first show in this subsection a loose but intuitive schematic (Fig. 4b) of our scenario, and then confirm it in the next subsection by numerical simulations.

For easier understanding, we first switch off the perpendicular propagation of the KAWs. The solid curves in Fig. 4b illustrate the trajectories of protons in this situation. For protons of small gyro-radius ($k_\perp \rho_i \ll 1$; blue curves), the spatial gradient-associated (i.e., $k_\perp$-associated) variations of wave phases (illustrated by the red-blue background in Fig. 4b) over the gyration period are small. The dynamics of these protons are approximately determined by the temporal variations[40,41], as in the case of electromagnetic ion cyclotron waves. These protons are well coupled to the KAWs' electric fields ($\mathbf{E_w}$), manifesting as a series of inclined and continuous stripes occurring periodically at the gyrophase of $\mathbf{E_w B_0}$[40,41] in gyrophase-time spectrograms.

In contrast, the spatial gradient plays a non-negligible role in the dynamics of protons of large gyro-radius ($k_\perp \rho_i \gtrsim 1$). As shown by the red solid curves, these protons would sample very different wave phases and experience oppositely directed $\mathbf{E_w}$ in the half-gyration period

before their detection. In this situation, work done by positive $\mathbf{E_w}$ roughly cancels out the work done by negative $\mathbf{E_w}$, resulting in very small net changes in proton energy and thus PSDs. Consequently, PBS gaps form.

To understand why PBS gaps are centered at the about 90° gyrophase, we now switch on the perpendicular propagation of the KAWs. The red dashed curves in Fig. 4b show the trajectories of protons of large gyro-radius in a frame of reference that moves at the perpendicular phase velocity of the waves in this situation. A comparison between the left and right columns indicates that, because of the relative motion between protons and waves, the wave phases sampled by the so-called trailing protons (which form 270°-gyrophase PBSs) vary more slowly than those sampled by the so-called head-on protons (which form 90°-gyrophase PBSs). Along their orbit, trailing protons only see the blue region and red region once each. In contrast, head-on protons first see the red region, then the blue region, next the red region, and finally the blue region again. Therefore, the effects of the spatial gradient are more significant for trailing protons than head-on protons, indicating the occurrence of the PBS gaps begins at the instant when the orbit of protons making up PBSs transitions to being head-on, i.e., at 90° gyrophase.

## Test particle simulations of KAW-proton interaction

We now employ test-particle simulations to illustrate proton dynamics in the fields of KAWs. Idealized wave fields (Fig. 5a) in the

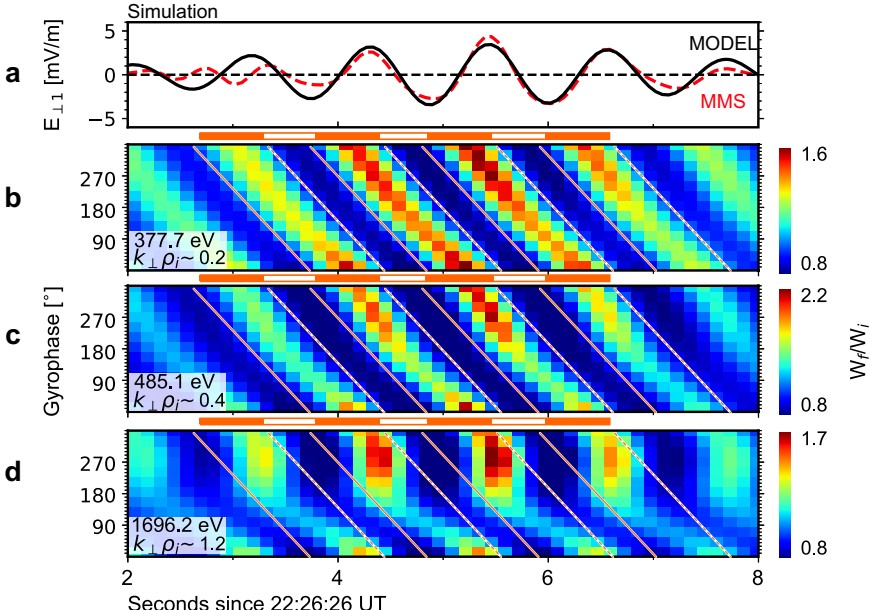

**Fig. 5 | Phase-bunching stripe gaps reproduced in numerical simulations. a** The modeled (black) and observed (red) electric fields of the KAWs. **b–d** The ratio between the final and initial proton energy, which is a proxy for the evolution of proton phase space densities. The orange solid and dashed lines give the expected positions of phase-bunching stripes. Phase-bunching stripes (marked by the orange lines and the bar below panel a) and phase-bunching stripe gaps in agreement with the observations are reproduced.

simulation are constrained using parameters derived from the observations, including the wave frequency, wavevector, and time profiles (see the Methods section for more information). We trace protons backward in time along their unperturbed trajectories by solving the equations of motion without the effects of wave modulation. Then, we integrate the work done by the wave electric field $\mathbf{E_w}$ along the unperturbed trajectory and obtain the initial energy ($W_i$) of protons before they enter the wave fields. The ratio between the final energy $W_f$ (the energy at which the backward trace starts) and $W_i$ is a proxy for the evolution of proton PSDs, since in this event the background PSDs generally decrease with increasing energy in the energy range of interest. Figures 5b–d shows the obtained $W_f/W_i$ distributions in the same format as Figs. 3e–g. To enable direct comparison between the two figures, the parallel velocity of protons in the simulation is set as −250 km/s, the same as the observations. PBSs and PBS gaps are reasonably well reproduced by the simulation. The PBS gaps center at 90° gyrophase and are more significant at higher energy in agreement with the observations. To elucidate the role of the parameter $k_\perp \rho_i$, we also conduct a controlled simulation with all parameters held constant except for the perpendicular wavelength, which is made much larger. As shown in Supplementary Fig. 4, continuous PBSs are observed in all energy channels of interest; no PBS gaps are observed.

Before closing this subsection, we return to the unusual oscillations observed in the pitch angle-time spectrograms (Figs. 3b–d). It is clear now that these oscillations result from the finite Larmor effects and PBS gaps: Because the gyrophase-averaged PSDs during PBS gaps are smaller than the gyrophase-averaged PSDs at other times, oscillations in the pitch angle-time spectrograms result. The resulting PSD oscillations should be larger at higher energy since the PBS gaps are more significant at higher energy.

**Sounding perpendicular spatial scale with particle gyration observations**

The above analysis provides a new way to derive the perpendicular wavelength $\lambda_\perp$ of KAWs from the observations of proton gyrophase distributions: The gyro-radius at which PBS gaps occur gives an estimate of the value of $\lambda_\perp$, which can be explicitly expressed as

$$\lambda_\perp = \frac{\sqrt{2 m_p E_u}}{e B_0} \sin \alpha, \tag{1}$$

where $E_u$ represents the highest energy channel where PBSs can be identified, $\alpha$ denotes the pitch-angle where PBSs are clearest, and $m_p$ and $e$ represent proton mass and charge, respectively. Further, as the (opposite) perpendicular propagation direction is given by the center gyrophase of PBS gaps, the perpendicular propagation of KAWs can actually be completely determined. Besides KAWs, this method, which is called particle-sounding technique here, can be applied to any other wave modes, as long as particles with gyro-radius comparable to corresponding $\lambda_\perp$ are observed.

In principle, the particle-sounding technique can only give a loose estimate of $\lambda_\perp$, as the transition from full PBSs to gapped PBSs is not very sharp. However, at least for the energy resolution of MMS/FPI, the last energy channel at which PBSs can be identified always give a reasonable estimate, as illustrated by the above analysis (Fig. 4a). This is understandable. Regarding particles with gyro-radius greater than $\lambda_\perp$, the work done by wave electric field would cancel out and thus no wave modulation is expected. Therefore, to apply the particle-sounding technique, one just needs to find the highest energy channel showing PBSs.

With the particle-sounding technique, we identified fifteen quasi-monochromatic wave events from the MMS dataset (including the one analyzed above), all of which are ion-scale, support significant parallel electric fields and propagate quasi-perpendicularly with respect to the background magnetic fields. Therefore, these waves are most likely KAWs. Supplementary Table 1 gives information about these events, including $\lambda_\perp$. This table also shows the ratio of $\lambda_\perp$ over MMS spacecraft separation ($R$). In about half of these events, $\lambda_\perp/R$ is larger than 10, indicating the timing analysis, which is one of the most popular methods for determining wavelength in the literature, may introduce large errors. Actually, a general trend is observed that the differences between the results of the particle-sounding technique and the results of the timing analysis increases as $\lambda_\perp/R$ increases. Figure 6 shows the

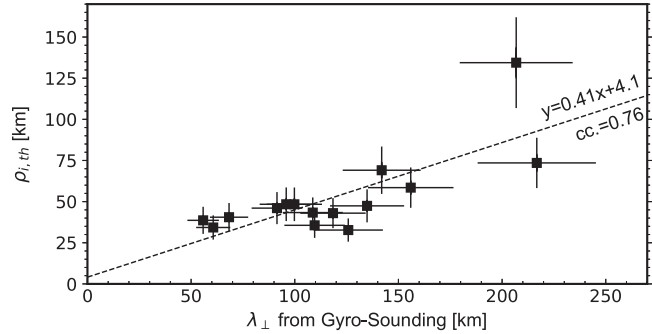

**Fig. 6 | The perpendicular wavelength of fifteen quasi-monochromatic KAW events observed by MMS.** The horizontal axis corresponds to the perpendicular wavelength derived from the particle-sounding technique, whereas the vertical axis represents proton gyro-radius evaluated at local proton perpendicular thermal energy $T_\perp$ (taken from the MMS/FPI/DIS-MOMS dataset). The dashed line shows a linear regression of the data, with the results denoted in the figure.

obtained $\lambda_\perp$ as a function of proton thermal gyro-radius defined as

$$\rho_{i,th} = \frac{\sqrt{2m_p T_\perp}}{eB_0},\qquad(2)$$

where $T_\perp$ represents proton perpendicular thermal energy (the error in $\lambda_\perp$ and $\rho_{i,th}$ is estimated as $\lambda_\perp \delta E_u / 2E_u$ and $\rho_{i,th}\delta T_\perp / 2T_\perp$, respectively, where $\delta E_u$ is the width of the corresponding energy channel and $\delta T_\perp$ represents the standard deviation of $T_\perp$ during the time interval when KAWs are observed). A good correlation is found: $\lambda_\perp \approx (2.4 \pm 0.7)\rho_{i,th}$, providing unambiguous evidence of the ion-scale nature of these KAWs.

## Discussion

In summary, we have systematically studied ion kinetics in KAWs based on MMS spacecraft observations. At small gyro-radius ($\rho_i \ll \lambda_\perp$), continuous PBSs indicative of nonlinear gyrophase trapping are observed. In contrast, at large gyro-radius ($\rho_i \gtrsim \lambda_\perp$), PBSs are not continuous, but gaps are observed in the direction opposite to the wave perpendicular propagation. Based on qualitative analysis and test-particle numerical simulation, these PBS gaps are attributed to KAW-ion decoupling induced by the finite Larmor gyro-radius effect at $\rho_i \sim \lambda_\perp$. We note the decoupling occurs at the energy corresponding to the local proton thermal energy, suggesting the bulk of the proton population is partially decoupled from the wave fields. This, together with the fact that electrons of much smaller mass remain frozen-in to the wave fields, experimentally demonstrates a theoretical expectation that KAWs arise from the decoupling of ion motions from electron motions[3].

The analysis reported here enables a practical method for measuring the perpendicular wavelength of KAWs. As discussed, the occurrence of PBS gaps at $\rho_i \sim \lambda_\perp$ is an intrinsic and observational feature of KAW-particle interaction. Therefore, when such PBS gaps are present, the corresponding gyro-radius quickly gives an estimate of $\lambda_\perp$. The methodology of deriving wave field spatial scales from particle-gyration measurements can be classified as a particle-sounding technique[42]. This technique is rarely used in wave studies whereas it has many advantages over other methods. First of all, it only requires data from a single spacecraft, while other methods like the timing and k-filtering techniques generally require coordination of at least four spacecraft. Secondly, the inputs required by the particle-sounding technique are relatively easy to obtain. In particular, unlike many other single-spacecraft methods, the particle-sounding technique does not require information on the full spectrum of particles. Instead, it only needs information on warm or hot (100 eV–10 keV) ions, which can be easily measured by, for example, electrostatic analyzers (ESAs). Third, the wavelength range for which the particle-

sounding technique is applicable is wide. For example, by providing a 50-nT background magnetic field and an ESA-type instrument with an energy range of 10 eV–10 keV, the particle-sounding technique is capable of simultaneously measuring perpendicular wavelength spanning from about 10 km to 300 km. In addition, we note the particle-sounding technique is still useful even when other methods are available, since it is based on very different measuring principles and thus can be used as an independent method to confirm the results from other methods.

However, it is necessary to point out that the particle-sounding technique requires particle gyrophase distributions with high time resolution as input. As a rule of thumb, the time resolution of particle instruments should be less than one-fifth of the wave periods to successfully apply this technique. Due to this limitation, at present, it is difficult to apply this technique to KAWs in Earth's magnetosphere detected by instruments other than MMS/FPI.

By applying the particle-sounding technique, fifteen quasi-monochromatic KAW events are then identified from the MMS database. An analysis of them reveals their perpendicular wavelength $\lambda_\perp$ is about $2.4 \pm 0.7$ times local proton thermal gyro-radius $\rho_{i,th}$ (Fig. 6), confirming the ion-scale nature of KAWs. As suggested by both theory and simulations[6-8], it is hard for coherent wave-particle interactions to produce particles with gyro-radius larger than the perpendicular wavelength of the waves involved, that is, $\sqrt{2m_p W_{\perp,m}}/eB_0 \lesssim \lambda_\perp$, where $W_{\perp,m}$ represents the highest perpendicular energy protons can reach. Therefore, the quantitative relationship obtained here also sets a strong intrinsic restriction on KAW-proton interactions: The highest perpendicular energy that protons can reach in coherent interactions with KAWs is 5.76 ($2.4^2$) times proton perpendicular thermal energy. Of course, this limit might be surpassed through stochastics[6-8], causing a few protons to reach higher energy. Nevertheless, it sets a restriction on the majority population.

## Methods
### Coordinate systems
Unless otherwise stated, the observations are presented in the field-aligned coordinate (FAC) system defined according to the background magnetic fields averaged over 22:26:26-22:26:36 UT, with the ∥-axis parallel to the background magnetic fields (increasing along the fields), the ⊥ 2-axis perpendicular to the plane given by the Sun-Earth line and the ∥-axis (increasing eastward), and the ⊥ 1-axis completing the right-handed coordinate system (increasing sunward). Particle pitch angle and gyrophase angle respecting to the background magnetic fields are then defined as the polar angle and azimuthal angle in this coordinate system, respectively. As defined in this way, the sunward direction corresponds to zero gyrophase. We note that, in the December 30, 2015, event, the propagation direction of the KAWs in the plane perpendicular to the background magnetic fields is roughly pointing towards the 270° gyrophase, as suggested by the wavevector obtained by Gershman et al.[12].

Besides the FAC system, the Geocentric Solar Ecliptic (GSE) coordinate system is also used, where the x-axis points towards the Sun along the Sun-Earth line, the z-axis is oriented along the ecliptic north pole, and the y-axis completes the right-handed coordinate system.

### Data processing
All displayed data in this manuscript are taken from MMS-1. Throughout the study, burst mode data are used.

The magnetic fields and electric fields were measured by the Fluxgate Magnetometers (FGM) instruments[43] and the Electric Field Double Probe (EDP) instruments[44,45], respectively. The frequency of the KAWs, about 0.88 Hz, was derived from a fast Fourier transform (Supplementary Fig. 1c). We employed a 0.5–2 Hz band-pass fifth-order Butterworth filter to separate the wave fields from the background

fields. The resulting wave fields are then transformed into the FAC system.

Ions were measured by the Fast Plasma Investigation (FPI)-Dual Ion Spectrometers (DIS) instruments[32]. Although the FPI-DIS instruments themselves do not distinguish between different ion species, the Hot Plasma Composition Analyzer (HPCA) instruments[33], which are capable of distinguishing minor ions (He$^+$, He$^{++}$ and O$+$) from protons, suggest the bulk of the ions detected by the FPI-DIS instruments are protons (approximately 99%). Thus, for convenience, we directly call FPI-DIS measurements protons. The FPI-DIS instruments provide a measurement of proton phase space density (PSD) velocity distributions every 150 milliseconds (ms) in its burst mode. This sampling time is only 10% of the periods of the KAWs, indicating the time resolution is high enough for our scope. The FPI-DIS instruments measure protons from 2.2 eV to 20 keV with 32 energy channels. The errors in the PSD data can be estimated based on the measurements from the four MMS satellites, which are equivalent to a set of repeated measures except for a time lag. The results suggest the relative standard errors are about 5%.

When generating Fig. 3a and Supplementary Fig. 2, a superposed epoch analysis was employed. First, five successive wave cycles were isolated from 22:26:29-22:26:35 UT. Then, the gyrophase distributions during the five wave cycles were linearly interpolated to make a uniformly spaced time series. This standard time series, which is called "wave phase", ranges from 0° to 360° and is evenly divided into eight bins. Finally, the five resulting distributions were superposed according to the wave phase. In this way, we obtained the so-called gyro-phase-wave phase spectrograms shown in Supplementary Fig. 2.

### Simulating the phase-bunching stripe gaps

Here, we employ a test-particle simulation to intuitively illustrate proton dynamics in the KAWs and the formation of the PBS gaps. In the simulations, the wave electric fields are modeled as

$$\begin{cases} E_{\perp 1} = -\exp\left[(t-t_0)^2/\tau^2\right] \times E_\perp \sin\phi \\ E_{\perp 2} = \exp\left[(t-t_0)^2/\tau^2\right] \times E_\perp \cos\phi \\ E_\parallel = -\exp\left[(t-t_0)^2/\tau^2\right] \times E_\parallel \sin\phi \end{cases}, \quad (3)$$

where $\phi = -\omega t + k_\parallel z + k_\perp y + \phi_0$. The results of the simulations are insensitive to wave parameters except the wavevector. Nevertheless, here we set all wave parameters according to the observations: $E_\perp = 3$ mV/m, $E_\parallel = 0.5$ mV/m, $k_\perp = 3.0 \times 10^{-3}$ rad/km ($3.0 \times 10^{-5}$ rad/km in the controlled simulation), $k_\parallel = 7.6 \times 10^{-3}$ rad/km, $\omega = 5.5$ rad/s, and $\tau = 3$ s. The factor $\exp[(t-t_0)^2/\tau^2]$ is involved to mimic the observed temporal profile of wave amplitude. In principle, wave amplitude also depends on spatial coordinates. However, since an unperturbed-trajectory approximation is used and all protons in the simulations have the same parallel velocity ($v_z$), any spatial variation in wave amplitude can be rewritten in a temporal form with a linear transformation $(z - z_0) = (\omega/k_\parallel - v_z)(t - t_0)$. Thus, for simplicity, only a temporal factor is used here.

As mentioned above, here we take an unperturbed-trajectory approximation. The basic idea of this approximation is that the effects of wave fields on proton motion are weak compared to the background fields. Therefore, to a first-order approximation, we can neglect the effects of wave fields when calculating proton motion, and then evaluate the role of wave fields by computing the work done by wave electric fields.

In the simulations, the background magnetic fields are modeled as a 55-nT uniform static field, whereas the background electric fields are set as zero everywhere. Therefore, the unperturbed proton velocity is simply a superposition of Larmor gyration in the plane perpendicular to the background magnetic fields and uniform motion in the parallel

direction. To compare the simulations with the observations presented in Figs. 3e–g, the parallel velocity of protons in the simulations is set as −250 km/s. Further tests suggest the overall results of the simulations are not sensitive to the specific value of proton parallel velocity.

After determining proton motion, we define proton energy gain as

$$\delta W = \int_{t_f - 100\tau}^{t_f} e\mathbf{v}(W, \alpha, \theta, t) \cdot \mathbf{E_w}(t) dt, \quad (4)$$

where $W$, $\alpha$ and $\theta$ are proton energy, pitch angle and gyrophase, respectively, $\mathbf{v}$ is proton velocities along the unperturbed trajectory, $\mathbf{E_w} = (E_{\perp 1}, E_{\perp 2}, E_\parallel)$ is the modeled wave electric field vector given in Eq. (3), and $t_f$ is the time at which the observations made (i.e., the time at which the backward trace starts). This integration can be done easily by standard numerical integration tools (see the code availability statement). Finally, we obtain the ratio between the final and initial energy as $W_f/W_i = W/(W - \delta W)$.

## Data availability

Data used in this study are archived at MMS Science Data Center (https://lasp.colorado.edu/mms/sdc/public), including magnetic field data (https://lasp.colorado.edu/mms/sdc/public/data/mms1/fgm/brst/l2/), electric field data (https://lasp.colorado.edu/mms/sdc/public/data/mms1/edp/brst/l2/dce/) and ion data (https://lasp.colorado.edu/mms/sdc/public/data/mms1/fpi/brst/l2/). The datasets generated during and/or analyzed during the current study are available from the corresponding author upon request.

## Code availability

MMS data have been loaded, analyzed, and plotted using the SpacePy package[46] for Python, which can be downloaded via the https://spacepy.github.io/install.html Installation page. Because of the unperturbed-trajectory approximation, the numerical simulations presented in this study reduce to integrating Eq. (4) numerically, which can be done by any standard numerical integration codes, for example, the Python function scipy.integrate (which can be downloaded via the https://scipy.org/install/ page).

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

## Acknowledgements

This work was supported by the National Natural Science Foundation of China 42230202 (Q.G.Z.), the Major Project of Chinese National Programs for Fundamental Research and Development 2021YFA0718600 (Q.G.Z.) and the China Space Agency project D020301 (Q.G.Z). We are grateful to the MMS team for providing the fields and plasma data. Thanks to Xu-Zhi Zhou and Shan Wang from Peking University for useful discussion.

## Author contributions

Z.Y.L. conducted the study, analyzed the data and prepared the manuscript. Q.G.Z. supervised the study, analyzed the data and revised the manuscript. Y.X.H. and S.Y.F. contributed to data analysis. R.R., H.Z., J.S.H., H.H.W. and C.Y. contributed to data analysis and revised the manuscript. C.J.P. and G.L. contributed to data processing of FPI-DIS and FGM, respectively.

## Competing interests

The authors declare no competing interests.
