## [Peer Review File · Nature Communications]

REVIEWER COMMENTS

Reviewer #1 (Remarks to the Author):

This paper presents a novel technique for measuring the perpendicular wavelength of kinetic Alfvén waves in space plasmas. The technique relies on the phase-bunching of ions gyrating in the presence of the wave. This is a potentially useful technique that merits eventual publication. However, there are a number of confusing points that should be clarified by the manuscript can be acceptable for publication.

The basic issue is that the authors need to be more clear on how the gaps in the phase space stripes are formed. One problem is that for some reason, they shift the point of 0 phase for Figure 4a compared with the other figures, including Figures 4b and 4c. It would be better if the same convention was used for all the figures. As Figures 4b and 4c show, and as is stated in the text, the gaps occur at about 90° gyrophase, where 0° is taken to be the sunward direction (this information is only found in the supplemental material; the definition of the phase should really be in the main text of the paper). If I am reading this right, the important point is that the gap is biggest for particles moving opposite to the phase velocity of the wave (the “head-on” particles) since these particles are spending some of their gyro-orbit in a region where the wave electric field is reversed, as described in lines 210-214 of the text. What I don’t understand is why there wouldn’t be a similar gap for the trailing particles if they had drawn a Figure like 4c at a time $t_0 + \Delta t$. One way to clarify this might be to plot the phase space trajectories in the plane of gyrophase vs. perpendicular velocity, which could clarify which particles are participating in the phase-trapping.

Another point is that the authors ignore other relevant work on the perpendicular wavelength of kinetic Alfvén waves. Spacecraft observations based on the E/B ratio can be used to verify the kinetic Alfvén wave dispersion relation, as was done, for example, by Stasiewicz et al. (GRL, 2000) and Chaston et al. (JGR, 2006). These observations make use of the rapid motion of satellites with respect to the plasma to determine the Doppler shift, in addition to interferometric techniques between the different probes on the spacecraft to determine the wavelength. Other important work comes from laboratory experiments at the Large Plasma Device at UCLA (Kletzing et al., PRL, 2010; Schroeder et al., Nature Comm., 2021). These works should be cited in the introduction.

One final note is that I would suggest modifying the color bar in Figures 2a and 2b. Everything below 700 eV or so is colored red and so it is difficult to pull out the oscillations that are observed.

In summary, this work could be potentially important, but the authors must clarify the process of phase trapping in order for it to be a valuable contribution to the literature.

Reviewer #2 (Remarks to the Author):

Review of the manuscript "Particle-sounding of the Spatial Structure of Kinetic Alfvén Waves" by Liu and coauthors

The manuscript presents an observational study of Kinetic Alfvén Waves, measured by the NASA MMS mission. The authors present a new technique to study the fluctuations associated with the waves, so that they can quantitatively determine the perpendicular wavelength of the Kinetic Alfvén Waves. With this information the authors estimate an upper bound for the energy that plasma particles can obtain due to resonant interactions with the waves.

Reported results seem relevant for plasma physics, particularly for the understanding of collisionless wave-particle interactions, and the role of Kinetic Alfvén Waves in space plasma processes. In general, the manuscript is not difficult to read. However, there are some issues that authors should clarify. I consider the paper can be published after the authors adequately answer and address the following comments:

Main issues

1. The main point of the article is to determine physical properties of Kinetic Alfvén Waves (KAWs) using observations. However, in order to do so, authors have to clarify that the observed waves are indeed KAWs, i.e., that they follow the KAWs dispersion relation. As this is not always possible with spacecraft observation, in the past several properties have been analyzed. As authors mention, the perpendicular wavelength similar to the ions gyroradius is an important feature of KAWs, but is not the only one. Determine properties such as a quasi-perpendicular wave-normal angle. the large parallel electric field and the signature right-handed polarization are also necessary in order to state that certain fluctuations correspond to KAWs (see e.g. Gary, 1986, *J. Plasma Phys.*; Wygant et al., 2002, *J. Geophys. Res.*; Chaston et al., 2014, *Geophys. Res. Lett.*; Moya et al., 2015, *J. Geophys. Res.*). I understand the authors based their analysis on Gershman et al. (2017) results but some clarifications are necessary.

2. How do authors know that the number density of ions is at least two orders of magnitude less than protons? Even though the HPCA instruments can provide an idea for the hot plasma, the warm or thermal plasma can behave quite different. This is an important issue as several studies have shown that the presence of ions can introduce important changes to the physics of KAWs (Venugopaul et al., 2014, *Phys. Scripta*; Tamrakar et al., 2017, *Astrophys. Space Sci.*; Barik et al., 2019, *Phys. Plasma*; Moya et al., 2022, *Astrophys. J.*). Please clarify and include the mentioned references.

3. Authors mentioned that "The existence of phase-bunching stripes (PBSs) indicates a strong coupling between the waves and protons". Please explain and justify.

4. Regarding the applicability of the particle sounding technique, authors have presented the results with probably the best plasma particles instrument ever onboard a spacecraft. It is possible to use the same technique with other satellites? Please comment.

5. Similar to comment #1, in line 269 it is said that another 14 events were identified with this technique. I understand the technique provides a way to obtain the perpendicular wavelength of waves propagating at oblique angles with respect to the magnetic field, but, how do authors know that the waves are indeed KAWs?

6. The discussion Section appears to be a summary rather than a proper discussion. Please improve the section including more details on the relevance and novelty of the proposed technique. In addition, in the same section authors conclude that the limit for the perpendicular energy gain is $5.76 (2.4^2)$ times the proton perpendicular temperature. How do authors reach such a conclusion?

Minor comments

-Line 24

Please explain the origin of the $5.76 (2.4^2)$ factor.

-Line 24 and 311

Instead of perpendicular temperature, do authors mean perpendicular thermal energy?

-Line 54

What do authors mean with "actual observation"?

-Line 99 and Data processing

How do authors obtained the PSDs data? What are the specifications of the measurements in terms of energy, errors, cadence, etc? Please specify and cite the data sets used.

-Line 303

It is mentioned that 15 events were identified. However, in line 269 the number of events is 14. Please clarify.

Reviewer #3 (Remarks to the Author):

The paper presents a novel technique to identify the spatial structure of kinetic Alfvén waves (KAWs) with in-situ spacecraft observations of particles. Although the importance of the non-gyrotropic particle distributions on wave-particle interactions has been demonstrated in previous studies [e.g., Kitamura et al, Science, 2018], they apply it to extract the spatial information of waves. I find that conclusions are well supported by the observational data and theoretical considerations. Therefore, I recommend to publish this paper with minor revision. Some minor comments/suggestions are shown below.

(1) If possible, please give an interpretation of the energy (2000eV in line 166) at which the PBSs disappear.

(2) According to theoretical considerations shown in Figs. 4 and 5, amplitude of KAWs seems not to be significant for the occurrence of gaps in PBSs. Can the authors add wave amplitude to Table S1 for confirmation?

(3) Table S1: "Fig.5" should read "Fig.6".

(4) Line 102: "rad/s" should read "rad/km".

(5) Line 209: "rho" should read "rho_i".

(6) In Gershman et al [Nat. Comm., 2017], the local generation of observed KAWs is not fully explained. Can the authors give some views on the local generation of KAWs listed in Table S1?

(7) If the authors have carried out test particle simulations without assumptions used in the present runs (e.g., no magnetic fluctuation, same parallel velocity), it is better to mention about those results. If not, please give explanations or comments for assumptions used in the present test particle simulations.

Response to Reviewer #1:

This paper presents a novel technique for measuring the perpendicular wavelength of kinetic Alfvén waves in space plasmas. The technique relies on the phase-bunching of ions gyrating in the presence of the wave. This is a potentially useful technique that merits eventual publication. However, there are a number of confusing points that should be clarified by the manuscript can be acceptable for publication.

We are very grateful to the reviewer for his/her efforts in evaluating this paper, and sincerely appreciate the constructive comments. We have revised the manuscript carefully according to these comments. In particular, we have further clarified how the gaps in the phase-bunching stripes are formed and modified the figures as suggested. Please find details in the following letter and the revised manuscript.

(Please note that the line numbers are for the text with track changes.)

1. The basic issue is that the authors need to be more clear on how the gaps in the phase space stripes are formed. One problem is that for some reason, they shift the point of 0 phase for Figure 4a compared with the other figures, including Figures 4b and 4c. It would be better if the same convention was used for all the figures. As Figures 4b and 4c show, and as is stated in the text, the gaps occur at about 90° gyrophase, where 0° is taken to be the sunward direction (this information is only found in the supplemental material; the definition of the phase should really be in the main text of the paper). If I am reading this right, the important point is that the gap is biggest for particles moving opposite to the phase velocity of the wave (the “head-on” particles) since these particles are spending some of their gyro-orbit in a region where the wave electric field is reversed, as described in lines 210-214 of the text. What I don’t understand is why there wouldn’t be a similar gap for the trailing particles if they had drawn a Figure like 4c at a time $t_0 + \Delta t$. One way to clarify this might be to plot the phase space trajectories in the plane of gyrophase vs. perpendicular velocity, which could clarify which particles are participating in the phase-trapping.

We sincerely appreciate these great comments.

We have modified Fig. 4a according to the reviewer’s suggestion. Now the same convention is used for all figures. Besides, to better illustrate the geometric configuration, we added three labels on top of Fig. 4a, which indicate the gyrophase of the sunward

direction, the PBS gaps and the k_{\perp} direction, respectively.

The Methods section has now been moved to the main text (Please find it after the Discussion section). Please find lines 392-394 in the main text for the definition of gyrophase:

Particle pitch angle and gyrophase angle respecting to the background magnetic fields are then defined as the polar angle and azimuthal angle in this coordinate system, respectively.

As defined in this way, the sunward direction corresponds to zero gyrophase.

We have also clarified further the formation of the gaps in phase-bunching stripes (PBSs). First, we apologize for making an error in the original Fig. 4b and 4c. That is, we incorrectly showed the head-on protons (which form 90° -gyrophase PBSs) and trailing protons (which form 270° -gyrophase PBSs) at the same time. However, they should correspond to protons detected by the instruments at different times (e.g., trailing protons at 4.4 s and head-on protons at 4.9 s, Fig. 3e-3g; the time shift is about half wave period). We have corrected this error. Please see the revised Fig. 4b. In addition, to better show what happens, we have added to Fig. 4b the trajectories of protons looking in a frame of reference that moves at the perpendicular phase velocity of the waves.

In the following, we further elaborate on our explanation for the formation of PBS gaps, and then answer the questions about PBS gaps asked in the comments.

First, we note that protons considered here are not in resonance with the KAWs, since the resonance velocities for both the Landau resonance and the cyclotron resonance, ~ -700 km/s and ~ 0 km/s, respectively, are far from the parallel velocity of protons considered, ~ -250 km/s. Consequently, there is no or at least very little secular energy exchange between protons and waves.

Second, a gyration cycle can be divided into two halves. In the first half, protons move roughly opposite to the perpendicular phase velocity of the waves. This opposite motion indicates a large relative velocity between protons and waves. As a result, the wave phases sampled by protons change rapidly, and the work done by wave electric fields is rapidly smoothed out during this period of time. In contrast, in the other half of gyration, protons move roughly in the same direction as the waves. Thus, the wave phases sampled by protons change relatively slowly, leaving nonzero work during this period of time. (Of course, the nonzero work is still temporal; it will be canceled in the next half of gyration.)

Combining the above two points together, the key point of the formation of PBS gaps is that the gaps are biggest for protons that move opposite to the perpendicular phase velocity of the waves just before their detection. Recalling our definition of PBSs (see Fig. 3e-3g), trailing protons are in the second half of gyration just before they are detected and thus would experience net acceleration, indicating relatively high PSDs. On the other hand, head-on protons are in the first half of gyration just before they are detected. Correspondingly, the cumulative wave modulation is weak for them, indicating relatively small PSDs. Admittedly, the above discussion is only a kind of simplified or even oversimplified graphic explanation. The actual process is not only determined by what happens during protons' last half of gyration but also by longer history. Mathematically speaking, the energy change can be expressed as a sum of sine and cosine functions weighted by Bessel functions. What remains after integration is determined by the competition of these terms. Nevertheless, the numerical simulations taking all the effects into account support the above graphic explanation.

Regarding why there would not be a similar gap for the trailing protons, please note that the trailing protons and head-on protons are the same group of protons detected at different times. The trailing and head-on orbits actually describe two phases of gyration, rather than two groups of protons.

The manuscript has been revised accordingly. Please find the revisions in the subsection "KAW-proton interaction at ion scales" (lines 212-244).

2. Another point is that the authors ignore other relevant work on the perpendicular wavelength of kinetic Alfvén waves. Spacecraft observations based on the E/B ratio can be used to verify the kinetic Alfvén wave dispersion relation, as was done, for example, by Stasiewicz et al. (GRL, 2000) and Chaston et al. (JGR, 2006). These observations make use of the rapid motion of satellites with respect to the plasma to determine the Doppler shift, in addition to interferometric techniques between the different probes on the spacecraft to determine the wavelength. Other important work comes from laboratory experiments at the Large Plasma Device at UCLA (Kletzing et al., PRL, 2010; Schroeder et al., Nature Comm., 2021). These works should be cited in the introduction.

We sincerely appreciate the comments. The suggested papers have been cited. Please see lines 52-58

The last class of methods utilizes wave dispersion relation presenting wave frequency as a

function of wave number. One may first construct or verify the dispersion relation from the observed ratio of electric fields to magnetic fields, from which wavelength can be extracted^{14,15}. For the sake of completeness, we also note the wavelength can sometimes be derived directly from the spatial distributions of relevant variables, if available. This method is usually applied in, for example, the study of aurora imaging¹⁶ and laboratory experiments^{17,18}.

3. One final note is that I would suggest modifying the color bar in Figures 2a and 2b. Everything below 700 eV or so is colored red and so it is difficult to pull out the oscillations that are observed.

Thanks very much for identifying this issue. Now more appropriate color bars are adopted. For each color bar, the reddest and bluest colors represent the maximum and minimum PSDs shown in the corresponding figure, respectively. We also added an additional panel in Fig. 2 showing the line-version of Fig. 2b. From this new panel one can see the oscillations more clearly. Please find panel c of the revised Fig. 2.

In summary, this work could be potentially important, but the authors must clarify the process of phase trapping in order for it to be a valuable contribution to the literature.

Once again, we sincerely appreciate the reviewer's effort in reviewing this manuscript.

Response to Reviewer #2:

The manuscript presents an observational study of Kinetic Alfvén Waves, measured by the NASA MMS mission. The authors present a new technique to study the fluctuations associated with the waves, so that they can quantitatively determine the perpendicular wavelength of the Kinetic Alfvén Waves. With this information the authors estimate an upper bound for the energy that plasma particles can obtain due to resonant interactions with the waves.

Reported results seem relevant for plasma physics, particularly for the understanding of collisionless wave-particle interactions, and the role of Kinetic Alfvén Waves in space plasma processes. In general, the manuscript is not difficult to read. However, there are some issues that authors should clarify. I consider the paper can be published after the authors adequately answer and address the following comments:

We are very grateful to the reviewer for his/her efforts in evaluating this paper, and sincerely appreciate the constructive comments. We have revised the manuscript carefully according to these comments. In the updated manuscript, we provided more details about the studied waves (including wave normal angle and the amplitude of parallel electric field), clarified how we obtained the PSD data, revised the Discussion section to include more comments on the particle-sounding technique, and added the references mentioned by the reviewer to the manuscript. Please find details in the following letter and the revised manuscript.

(Please note that the line numbers are for the text with track changes.)

Main issues

1. The main point of the article is to determine physical properties of Kinetic Alfvén Waves (KAWs) using observations. However, in order to do so, authors have to clarify that the observed waves are indeed KAWs, i.e., that they follow the KAWs dispersion relation. As this is not always possible with spacecraft observation, in the past several properties have been analyzed. As authors mention, the perpendicular wavelength similar to the ions gyroradius is an important feature of KAWs, but is not the only one. Determine properties such as a quasi-perpendicular wave-normal angle, the large parallel electric field and the signature right-handed polarization are also necessary in order to state that certain fluctuations correspond to KAWs (see e.g. Gary, 1986, J.

Plasma Phys; Wygant et al., 2002, J. Geophys. Res; Chaston et al., 2014, Geophys. Res. Lett; Moya et al., 2015, J. Geophys. Res). I understand the authors based their analysis on Gershman et al. (2017) results but some clarifications are necessary.

We sincerely appreciate the comments. In the revised manuscript, we have added more details about the properties of the waves. We highlight the fact that the parallel component of the wave electric fields is significant, and the waves propagate quasi-perpendicularly to the background magnetic fields with a wave normal angle of $\sim 88.5^\circ$. These results further support the KAW-mode nature of the waves studied here. Please find the revisions in lines 127-132

However, significant oscillations can also be observed in the parallel component of the wave electric fields (E_{\parallel}); the amplitude of E_{\parallel} is about 0.7 mV/m, as shown in Fig. 2d. To determine wave propagation, a minimum variance analysis²⁹ is applied to the wave magnetic fields. The results show the waves propagate quasi-perpendicularly to \mathbf{B}_0 , with a wave normal angle of about 88.5° . The properties listed above support the identification of the waves as KAWs^{30,31}.

We note Gershman et al. (2017) have already conducted an analysis of the dispersion relation (see their Figure 5). This analysis is now quoted in our manuscript. Please find lines 111-112

From this, together with an analysis of the dispersion relation (their Figure 5), Gershman et al. concluded the KAW-mode nature of the observed waves.

2. How do authors know that the number density of ions is at least two orders of magnitude less than protons? Even though the HPCA instruments can provide an idea for the hot plasma, the warm or thermal plasma can behave quite different. This is an important issue as several studies have shown that the presence of ions can introduce important changes to the physics of KAWs (Venugopaul et al., 2014, Phys. Scripta; Tamrakar et al., 2017, Astrophys. Space Sci; Barik et al., 2019, Phys. Plasma; Moya et al., 2022, Astrophys J.). Please clarify and include the mentioned references.

We sincerely appreciate the comments. The statement on ion species is based on HPCA measurements. We totally agree with the reviewer that the HPCA measurements cannot fully rule out the presence of multi-ion effects in this event, which would significantly affect the physics of KAWs as suggested. We have made further clarification and cited the references listed above (ref. 34-36; thanks for the recommendation). However, we

suggest these effects do not really affect our analysis. For our purpose here, we only need to confirm what is observed by the FPI instruments is mostly protons, so that we can correctly calculate the gyro-radii of the detected ions. The HPCA instruments are able to do this, since they cover a very similar energy range as the FPI instruments. We have revised the manuscript accordingly in lines 136-143 as following:

Although the FPI instruments do not distinguish between different ion species, the HPCA instruments³³ on MMS, which cover a similar energy range as the FPI instruments, suggest the bulk of the ions detected by the FPI instruments are protons. Therefore, the gyro-radii of the ions detected by the FPI instruments can be calculated based on the proton mass and charge, although the multi-ion effects³⁴⁻³⁶ on KAW physics cannot be fully ruled out. The latter effects, however, are beyond the scope of this paper, and therefore, we will hereafter refer to the FPI ion measurements as proton measurements for convenience.

3. Authors mentioned that "The existence of phase-bunching stripes (PBSs) indicates a strong coupling between the waves and protons". Please explain and justify.

Sorry for this vague sentence. Here we simply want to say that the existence of repeated PBSs indicates protons are modulated by wave fields. After careful reconsideration, we think this sentence is not very related to the context and is a little bit confusing. Hence, we deleted it in the updated manuscript. Please see lines 173-174.

4. Regarding the applicability of the particle sounding technique, authors have presented the results with probably the best plasma particles instrument ever onboard a spacecraft. It is possible to use the same technique with other satellites? Please comment.

Thanks for the comments. Gyrophase distributions with appropriate time, energy and angle resolution are required to successfully apply the particle-sounding technique. The most challenging parameter is the time resolution, which should be less than one-fifth of wave periods, as a rule of thumb. Regarding KAWs in most regions of Earth's magnetosphere, the MMS/FPI instrument is the only one that meets this requirement so far.

We note that the requirement on the time resolution is easier to meet in regions of lower magnetic fields, since wave periods generally scale with particle gyro-frequency. For regions like central plasma sheet, foreshock and the interplanetary space, a time resolution of ~ 1 s is enough for applying the particle-sounding technique to many ion-

scale waves in these regions. Future space missions focusing on small-scale dynamics and cross-scale processes may take our technique into account when designing instruments.

We have added the above discussion to the Discussion section. Please see the response to major comment #6.

5. Similar to comment #1, in line 269 it is said that another 14 events were identified with this technique. I understand the technique provides a way to obtain the perpendicular wavelength of waves propagating at oblique angles with respect to the magnetic field, but, how do authors know that the waves are indeed KAWs?

Thanks for identifying this issue. Indeed, strictly speaking, we cannot fully confirm the waves are KAWs, since an analysis of the dispersion relation is not possible in practice for most of the cases listed here. However, we note many properties of the waves are consistent with our suggestion. The particle-sounding technique suggests the perpendicular wavelength of these waves is of the same order of magnitude as thermal proton gyro-radius. We have also added the wave parallel electric fields and wave normal angle in the revised Table S1. As shown, all waves listed here have significant parallel electric fields and propagate quasi-perpendicularly to the background magnetic field. Therefore, it should be reasonable to suggest all the waves are KAWs. We also revised the main text. Please find lines 316-319

With the particle-sounding technique, we identified fifteen quasi-monochromatic wave events from the MMS dataset (including the one analyzed above), all of which are ion-scale, support significant parallel electric fields and propagate quasi-perpendicularly with respect to the background magnetic fields. Therefore, these waves are most likely KAWs.

6. The discussion Section appears to be a summary rather than a proper discussion. Please improve the section including more details on the relevance and novelty of the proposed technique. In addition, in the same section authors conclude that the limit for the perpendicular energy gain is $5.76 (2.4^2)$ times the proton perpendicular temperature. How do authors reach such a conclusion?

We sincerely appreciate the reviewer for identifying these issues.

We have revised the Discussion section as suggested. We added the discussion about

the advantages and disadvantages of the particle-sounding technique in lines 352-371 as following:

This technique is rarely used in wave studies whereas it has many advantages over other methods. First of all, it only requires data from a single spacecraft, while other methods like the timing and k-filtering techniques generally require coordination of at least four spacecraft. Secondly, the inputs required by the particle-sounding technique are relatively easy to obtain. In particular, unlike many other single-spacecraft methods, the particle-sounding technique does not require information on the full spectrum of particles. Instead, it only needs information on warm or hot (~ 100 eV-10 keV) ions, which can be easily measured by, for example, electrostatic analyzers (ESAs). Third, the wavelength range for which the particle-sounding technique is applicable is wide. For example, by providing a 50-nT background magnetic field and an ESA-type instrument with an energy range of 10 eV-10 keV, the particle-sounding technique is capable of simultaneously measuring perpendicular wavelength spanning from about 10 km to 300 km. In addition, we note the particle-sounding technique is still useful even when other methods are available, since it is based on very different measuring principles and thus can be used as an independent method to confirm the results from other methods.

However, it is necessary to point out that the particle-sounding technique requires particle gyrophase distributions with high time resolution as input. As a rule of thumb, the time resolution of particle instruments should be less than one-fifth of the wave periods to successfully apply this technique. Due to this limitation, at present, it is difficult to apply this technique to KAWs in Earth's magnetosphere detected by instruments other than MMS/FPI.

The limit for the perpendicular energy gain is based on previous results that waves can only coherently accelerate ions up to a perpendicular energy (W_{\perp}) at which the ion gyro-radius evaluated ($\frac{\sqrt{2m_p W_{\perp,m}}}{eB_0}$) is roughly equal to the perpendicular wavelength of the waves (λ_{\perp}) [see, e.g., Lysak et al., 1980, Chaston et al., 2004 and Shen et al., 2020; all cited in the manuscript]. On the other hand, our results show $\lambda_{\perp} \sim 2.4\rho_{i,th}$, where $\rho_{i,th} = \frac{\sqrt{2m_p T_{\perp}}}{eB_0}$ and T_{\perp} represents proton perpendicular thermal energy. Therefore, we have $\frac{\sqrt{2m_p W_{\perp,m}}}{eB_0} \sim \lambda_{\perp} \sim 2.4 \frac{\sqrt{2m_p T_{\perp}}}{eB_0}$, giving $W_{\perp,m} \sim 2.4^2 T_{\perp} = 5.76 T_{\perp}$. In this way, we reach the conclusion on the limit for proton perpendicular energy gain. We have revised the manuscript to make this point more clearly in lines 376-382 as following:

As suggested by both theory and simulations⁶⁻⁸, it is hard for coherent wave-particle

interactions to produce particles with gyro-radius larger than the perpendicular wavelength of the waves involved, that is, $\frac{\sqrt{2m_p W_{\perp,m}}}{eB_0} \lesssim \lambda_{\perp}$, where $W_{\perp,m}$ represents the highest perpendicular energy protons can reach. Therefore, the quantitative relationship obtained here also sets a strong intrinsic restriction on KAW-proton interactions: The highest perpendicular energy that protons can reach in coherent interactions with KAWs is 5.76 (2.4^2) times proton perpendicular thermal energy.

Minor comments

1. Line 24: Please explain the origin of the 5.76 (2.4^2) factor.

Please see the response to major comment #6.

2. Line 24 and 311: Instead of perpendicular temperature, do authors mean perpendicular thermal energy?

Yes, we mean “perpendicular thermal energy”. We have replaced all “perpendicular temperature” in the manuscript with “perpendicular thermal energy”. Please see lines 25, 78, 327, 338, 382 and 683.

3. Line 54: What do authors mean with "actual observation"?

Sorry for the confusion. “Actual observation” refers to what is cited in the following sentences in the same paraphrase. We have deleted this confusing phrase. Please see line 61 for the revision.

4. Line 99 and Data processing: How do authors obtained the PSDs data? What are the specifications of the measurements in terms of energy, errors, cadence, etc? Please specify and cite the data sets used.

The PSD data is taken from the FPI-DIS instruments. The revised manuscript provides more details about the FPI-DIS instruments in lines 403-425 as following:

All displayed data in this manuscript are taken from MMS-1...Throughout the study, burst mode data are used...Ions were measured by the Fast Plasma Investigation (FPI)-Dual Ion Spectrometers (DIS) instruments...The FPI-DIS instruments provide a measurement of

proton phase space density (PSD) velocity distributions every 150 milliseconds (ms) in its burst mode... The FPI-DIS instruments measure protons from ~2.2 eV to 20 keV with 32 energy channels. The errors in the PSD data can be estimated based on the measurements from the four MMS satellites, which are equivalent to a set of repeated measures except for a time lag. The results suggest the relative standard errors are about 5%.

Also, please find in the Data Availability section the links for the data, including the FPI-DIS PSD data, in lines 474-480 as following:

MMS data used in this study are archived at MMS Science Data Center (<https://lasp.colorado.edu/mms/sdc/public>), including magnetic field data (<https://lasp.colorado.edu/mms/sdc/public/data/mms1/fgm/brst/12/>), electric field data (<https://lasp.colorado.edu/mms/sdc/public/data/mms1/edp/brst/12/dce/>) and ion data (<https://lasp.colorado.edu/mms/sdc/public/data/mms1/fpi/brst/12/>).

5. Line 303: It is mentioned that 15 events were identified. However, in line 269 the number of events is 14. Please clarify.

The 15 events mentioned in line 303 (for the old version) include the one analyzed in the main text, while the 14 events mentioned in line 209 (for the old version) do not. We have revised the manuscript to eliminate the confusion in lines 316-317 as following:

With the particle-sounding technique, we identified fifteen quasi-monochromatic wave events from the MMS dataset (including the one analyzed above)...

Once again, we sincerely appreciate the reviewer's effort in reviewing this manuscript.

Response to Reviewer #3:

The paper presents a novel technique to identify the spatial structure of kinetic Alfvén waves (KAWs) with in-situ spacecraft observations of particles. Although the importance of the non-gyrotropic particle distributions on wave-particle interactions has been demonstrated in previous studies [e.g., Kitamura et al, Science, 2018], they apply it to extract the spatial information of waves. I find that conclusions are well supported by the observational data and theoretical considerations. Therefore, I recommend to publish this paper with minor revision. Some minor comments/suggestions are shown below.

We are very grateful to the reviewer for his/her efforts in evaluating this paper, and sincerely appreciate the constructive comments. We have revised the manuscript carefully according to these comments. We especially elaborate on our assumptions used in the simulations. Please find details in the following letter and the revised manuscript.

(Please note that the line numbers are for the text with track changes.)

1. If possible, please give an interpretation of the energy (2000eV in line 166) at which the PBSs disappear.

Thanks. An explanation can be inferred from the subsection “KAW-proton interaction at ion scales” (lines 212-244). In general, as the perpendicular energy and thus gyro-radius of protons increase, the wave phase sampled by protons during gyration varies more significantly. As a result, the work done by wave electric fields would be smoothed out, leading to the disappearance of PBSs. Clearly, wave perpendicular wavelength controls the perpendicular energy at which PBSs disappear.

2. According to theoretical considerations shown in Figs. 4 and 5, amplitude of KAWs seems not to be significant for the occurrence of gaps in PBSs. Can the authors add wave amplitude to Table S1 for confirmation?

Thanks very much for the great suggestion. We have added the amplitude of wave perpendicular magnetic fields (B_{\perp}) to Table S1. B_{\perp} varies significantly from case to case, ranging from 1.0 nT to 5.1 nT for the 15 events listed in Table S1. This result indeed suggests that the amplitude of KAWs is not critical for PBS gaps formation.

3. Table S1: "Fig.5" should read "Fig.6".

Have revised. Please find the title of Table S1.

4. Line 102: "rad/s" should read "rad/km".

Thanks. Have revised. Please find line 109.

5. Line 209: "rho" should read "rho_i".

Have revised. Please find lines 227.

6. In Gershman et al [Nat. Comm., 2017], the local generation of observed KAWs is not fully explained. Can the authors give some views on the local generation of KAWs listed in Table S1?

The local generation of KAWs is an important issue. This is also one of our motivations for developing the particle-sounding technique. However, we have to admit that we cannot answer this question at this stage. Very preliminary results suggest the association between KAWs and ion beams. However, a detailed discussion of this point is much beyond the scope of this paper. Further studies on a large KAW dataset established by the particle-sounding technique may provide information on when, where and how KAWs occur and lead to a more comprehensive understanding of the local generation of KAWs.

7. If the authors have carried out test particle simulations without assumptions used in the present runs (e.g., no magnetic fluctuation, same parallel velocity), it is better to mention about those results. If not, please give explanations or comments for assumptions used in the present test particle simulations.

Thanks very much for the comment. We did not carry out any other test particle simulations.

The usage of the same parallel velocity is actually not based on an assumption. We adopt it just because Fig. 3e-3g, which present the observed gyrophase distributions, are made for protons with the same parallel velocity ~ -250 km/s that is around the background jet velocity. We fixed the parallel velocity of protons (also -250 km/s) in the

simulations, so that the results of the simulations can be directly compared with the observations. Now, we have stated this point explicitly in lines 282-285:

Fig. 5b-5d shows the obtained W_f/W_i distributions in the same format as Fig. 3e-3g. To enable direct comparison between the two figures, the parallel velocity of protons in the simulation is set as -250 km/s , the same as the observations..

and lines 457-462:

In the simulations, the background magnetic fields are modeled as a 55-nT uniform static field, whereas the background electric fields are set as zero everywhere. Therefore, the unperturbed proton velocity is simply a superposition of Larmor gyration in the plane perpendicular to the background magnetic fields and uniform motion in the parallel direction. To compare the simulations with the observations presented in Fig. 3e-3g, the parallel velocity of protons in the simulations is set as -250 km/s .

We further note that the value of proton parallel velocity does not affect the simulations much. To illustrate this, the figure attached below shows the results of a simulation with the parallel velocity of protons setting as -200 km/s . One can see it is overall similar to Fig. 5 in the manuscript. It also shows relatively continuous PBSs at small $k_{\perp}\rho_i$ but PBS gaps at $k_{\perp}\rho_i \sim 1$. This point is added to the manuscript in lines 462-463 as following:

Further tests suggest the overall results of the simulations are not sensitive to the specific value of proton parallel velocity.

Figure R1. The same as Fig. 5, except proton parallel velocity is taken as -200 km/s .

Regarding the assumption of no magnetic fluctuation, we use it because here we take an unperturbed-trajectory approximation, which is widely used in the study of wave-particle interactions, e.g., Dungey (1963, <https://doi.org/10.1017/S0022112063000069>) and Southwood and Kivelson (1981, <https://doi.org/10.1029/JA086iA07p05643>). The basic idea of this approximation is that the effects of wave fields on proton motion are weak compared to the background field. Thus, to a first-order approximation, we can neglect wave modulation when calculating proton motion, and then evaluate the role of wave fields by computing the work done by wave electric fields. Since wave magnetic fields do not do any work, they do not appear in this procedure explicitly. We have added the above discussions to the manuscript in lines 450-454 as following:

As mentioned above, here we take an unperturbed-trajectory approximation. The basic idea of this approximation is that the effects of wave fields on proton motion are weak compared to the background fields. Therefore, to a first-order approximation, we can neglect the effects of wave fields when calculating proton motion, and then evaluate the role of wave fields by computing the work done by wave electric fields.

Once again, we sincerely appreciate the reviewer's effort in reviewing this manuscript.

REVIEWERS' COMMENTS

Reviewer #1 (Remarks to the Author):

The authors of this manuscript have responded positively to all of the comments in my first review, and in my opinion, to the comments of the other reviewers. I now think this is a noteworthy publication worthy of publication in Nature Communications.

Reviewer #2 (Remarks to the Author):

Second Review of the manuscript "Particle-sounding of the Spatial Structure of Kinetic Alfvén Waves" by Liu and coauthors.

I am satisfied with how the authors addressed my previous comments. I consider the paper can be published in Nature Communications.

Reviewer #3 (Remarks to the Author):

The authors have adequately addressed all my comments. I recommend the revised manuscript for publication.

Response to Reviewer #1:

The authors of this manuscript have responded positively to all of the comments in my first review, and in my opinion, to the comments of the other reviewers. I now think this is a noteworthy publication worthy of publication in Nature Communications.

We are very grateful to reviewer #1 for his/her continued efforts in evaluating this paper.

Response to Reviewer #2:

Second Review of the manuscript "Particle-sounding of the Spatial Structure of Kinetic Alfvén Waves" by Liu and coauthors.

I am satisfied with how the authors addressed my previous comments. I consider the paper can be published in Nature Communications.

We are very grateful to reviewer #2 for his/her continued efforts in evaluating this paper.

Response to Reviewer #3:

The authors have adequately addressed all my comments. I recommend the revised manuscript for publication.

We are very grateful to reviewer #3 for his/her continued efforts in evaluating this paper.